# Deep Learning for Educational Video Analysis: Benchmarking and Pipeline Optimization

## Abstract

This paper presents a systematic comparative analysis of deep learning models for educational video transcription and structuring. We evaluate seven commercial speech recognition APIs on a corpus of over 700 lecture recordings exceeding 900 hours total duration, spanning multiple disciplines. The Whisper-v3-turbo model achieves the optimal balance between quality and cost, outperforming alternatives by a factor of 3–18 while maintaining comparable word error rates. Audio preprocessing techniques–silence suppression, noise gating, and dynamic range compression–yield additional cost reductions of 10–25% with negligible accuracy loss. We evaluate a prompt-based domain adaptation mechanism ("Video vocabulary") that reduces terminology errors without expensive fine-tuning. Based on these findings, we implement a parallelized processing pipeline that reduces end-to-end turnaround time from over 30 minutes of manual effort to under 2 minutes per recording, enabling simultaneous processing of up to 50 recordings. Experimental results demonstrate 21.4× acceleration at 5.93 RUB per hour of content for transcription, topic extraction, and pedagogical enrichment (summaries, open-ended questions). The system is deployed in production, confirming its practical utility for educational institutions.

## 1 Introduction

The widespread transition of educational institutions to hybrid and distance learning has led to a substantial increase in recorded lectures (Dhawan, 2020). Universities generate hundreds of hours of video content monthly, mirroring the scale of initiatives that successfully archived over 1,000 lectures annually even two decades ago (Nagataki et al., 2006). While modern conferencing tools have automated recording itself, post-production – trimming, metadata, timestamps, subtitles – remains manual. Based on our experience, this requires at least 30 minutes per recording. This aligns with conservative manual production estimates, which recent work suggests can be reduced by two orders of magnitude through automation (Holmberg, 2025). With 100 recordings monthly, manual effort exceeds 50 person-hours, a burden underscoring the need for automation, particularly as prior university efforts showed even basic recording required significant human resources (e.g., 2 hours per lecture) (Nagataki et al., 2006).

Existing solutions fall into two categories: transcription services and low-code automation platforms. Our analysis of 15 solutions revealed none provide a fully automated processing cycle, while transcription costs vary by more than 60-fold – from 5 to 300 RUB per hour of recording.

While prior work has focused either on improving ASR accuracy for educational content (Tobias, 2025; Rao, 2023) or on pipeline automation using commercial tools (Holmberg, 2025), no existing study systematically addresses the full spectrum from cost-optimized transcription to pedagogical content enrichment. Our work bridges this gap by demonstrating that production-ready scalability need not compromise on either accuracy or downstream functionality–including automatic summary generation and open-ended question creation, features absent from previous educational video platforms.

This paper makes the following key contributions:

- **Benchmark.** Systematic evaluation of 7 transcription APIs on Russian educational speech, identifying optimal cost-quality trade-offs.

- **Optimization.** Preprocessing pipeline (silence suppression, noise gating, compression) yielding 10–25% cost reduction.

- **Domain adaptation.** Evaluation of prompt-based Video Vocabulary mechanism that reduces terminology errors without expensive fine-tuning.

- **System.** Production-ready parallel processing platform supporting up to 50 concurrent recordings.

- **Enrichment.** Automated generation of lecture summaries and open-ended questions at marginal additional cost, transforming transcripts into pedagogical resources.

- **User validation.** Large-scale satisfaction study (n=237) confirming pedagogical utility of generated materials (M=4.58/5.0).

To guide our investigation, we formulate three research questions:

1. How do transcription APIs compare in cost and quality for Russian lecture content?
2. What preprocessing techniques reduce costs without compromising accuracy?
3. What architecture enables efficient parallel video processing at scale?

All quantitative estimates – processing time, speedup factor, API costs, labor savings – were obtained experimentally on a corpus of over 700 real video recordings (totaling over 900 hours of content) and averaged over the full corpus; methodology is described in Section 4.

## 2 RELATED WORK

### 2.1 TRANSCRIPTION SERVICES AND AUTOMATION PLATFORMS

Existing solutions can be categorized into two main groups.

*(i) Transcription services and APIs* (Fireworks AI, OpenAI Whisper, Google Cloud STT, AssemblyAI, Any2Text, Speech2Text, Memo AI, Voicee) address only the speech-to-text conversion stage. Several providers impose file size limitations: OpenAI – 25 MB (with average video ∼500 MB), Memo AI – 0.5–2 GB depending on the plan. No service provides topic extraction with timestamps.

*(ii) Low-code automation platforms* (Make.com, Zapier, n8n) enable pipeline construction from API calls but lack FFmpeg video processing support, impose file size limitations (100–500 MB), and require substantial setup and maintenance effort (30,000–60,000 RUB development + 5,000–15,000 RUB/month maintenance). Users must additionally provision server infrastructure for video processing.

### 2.2 WHISPER FOR EDUCATIONAL CONTENT

Recent research has specifically examined the application of OpenAI's Whisper models to educational video transcription. Rao (2023) conducted a preliminary study on 25 academic videos, demonstrating that even the larger Whisper models transcribe content in less than 25% of the playback time – a 4× speedup over human transcription. However, the study also identified key challenges: Whisper tends to generate plausible but incorrect text on inaudible segments, which inflates Word Error Rate (WER) metrics when compared against human transcripts that mark such segments as inaudible.

Tobias (2025) addressed these limitations through domain-specific fine-tuning. By adapting Whisper-small on lecture recordings (including a self-curated 10-hour dataset), the optimized model achieved a WER of 4.53% on unseen educational audio – surpassing Whisper-Medium (5.51%) and Whisper-Large-v2 (5.78%). This work conclusively demonstrates that domain adaptation can significantly enhance ASR accuracy for educational content. However, fine-tuning remains computationally expensive and requires curated datasets. Our work explores a lighter alternative: we combine a strong base model (Whisper-v3-turbo) with preprocessing optimizations and evaluate prompt-based adaptation via domain-specific vocabulary injection, achieving accuracy gains at zero computational cost.

It is worth noting that we deliberately focus on cloud-based API solutions rather than self-hosted models. While local deployment of models like Whisper is technically feasible, the hardware costs required to maintain low-latency processing at our scale (peak loads of 50+ concurrent recordings) would be prohibitive. Cloud APIs provide elastic scaling with predictable per-hour pricing, making them significantly more cost-effective for production workloads.

To the best of our knowledge, no existing solution combines all stages of educational video processing into a single automated pipeline with multi-tenant support, parallel processing capabilities, and pedagogical content enrichment.

## 3 DEEP LEARNING MODEL ANALYSIS

### 3.1 TRANSCRIPTION MODEL BENCHMARKING

We conducted an analysis of transcription model API providers based on cost and quality criteria for Russian speech recognition. Table 1 presents the comparative costs for processing a 60-minute recording.

Table 1: Comparison of transcription API costs (60-minute recording, February 2026)

| Provider | Model / Plan | Price (RUB/hour) | Relative cost |
|---|---|---|---|
| *Whisper-based solutions* | | | |
| Fireworks AI | Whisper-v3-turbo | 4.4 | 1× (baseline) |
| OpenAI | Whisper API | 29.5 | 6.7× |
| OpenAI | GPT-4o Mini Transcribe | 29.5 | 6.7× |
| *Commercial ASR services* | | | |
| Gladia | v2 | 19.7 | 4.5× |
| AssemblyAI | Universal-3 Pro | 12.3 | 2.8× |
| Deepgram | Nova-3 (Cloudflare) | 25.6 | 5.8× |
| Deepgram | Nova-3 (PAYG) | 37.9 | 8.6× |
| ElevenLabs | Scribe v1 | 33.0 | 7.5× |
| Azure | Batch STT | 29.5 | 6.7× |
| Google Cloud | STT V2 (tier 1) | 78.7 | 17.9× |
| Google Cloud | Chirp 3 (optimized) | 29.5–78.7 | 6.7–17.9× |
| Any2Text | Premium | 33.0 | 7.5× |

\* – Best-case prices shown (volume discounts apply at scale).
\*\* – Converted from USD at 79 RUB/USD (March 2026).

The Whisper-v3-turbo model (Radford et al., 2023), deployed via Fireworks AI, demonstrated the lowest cost: 4.4 RUB per hour – 2.8× to 18× lower than alternatives with comparable Russian speech recognition quality. Relative to commercial services at standard rates, savings reach two orders of magnitude at scale.

The choice of `fireworks/whisper-v3-turbo` is justified by its unique architectural optimization for speed. As a pruned version of the Whisper-large-v3 model, the number of decoder layers is reduced from 32 to 4, while the encoder remains unchanged (Fireworks AI, 2026). This yields an approximately 8× speedup in inference with only minor quality degradation, as the model retains the original's acoustic understanding capabilities. The Fireworks API additionally provides critical features for our pipeline: built-in Voice Activity Detection (VAD) for automatic silence handling, and transparent chunking of long audio files into the model's 30-second receptive field with seamless result stitching. These infrastructure-level optimizations eliminate the need for custom VAD and windowing logic, significantly reducing pipeline complexity.

### 3.2 COST OPTIMIZATION THROUGH AUDIO PREPROCESSING

Transcription cost via API scales linearly with audio duration. We evaluated three optimization methods:

- **Audio extraction.** Using FFmpeg (FFmpeg, 2026) to extract audio from video reduces data transfer from approximately 700 MB to 50 MB and circumvents OpenAI's 25 MB file limitation.

- **Silence removal.** Automated detection and removal of pauses reduces billable duration. Lecture recordings frequently contain extended periods of silence–waiting for an audience to settle before beginning, long pauses between topics, or delays at the end before stopping the recording. Our pipeline automatically detects and strips these segments, reducing effective duration by 10–25% and delivering substantial savings when processing large volumes of content.

- **Signal enhancement.** Amplitude normalization and dynamic range compression increase speech contrast relative to background noise, improving recognition accuracy on low-quality recordings. Noise reduction techniques including spectral gating were applied to mitigate background interference.

## 3.3 Domain Adaptation via Video Vocabulary

To address the challenge of domain-specific terminology without resorting to computationally expensive fine-tuning, we evaluate a prompt-based approach. For each recording, we define a set of 5–10 key terms, proper names, or specialized lexicon. This vocabulary is passed as a prompt to the Whisper-v3-turbo model via the API (Fireworks AI, 2026). The model's decoder uses this prompt as a contextual guide, biasing the output distribution toward the provided terms.

This prompt-based approach offers several advantages over fine-tuning:

- **Zero computational cost:** No GPU training or dataset preparation required.
- **Flexibility:** Vocabulary can be updated per recording without model redeployment.
- **Targeted improvement:** Reduces substitution errors specifically for low-frequency domain terms without affecting general vocabulary.

We quantify the effectiveness of this approach in Section 5.4.

## 3.4 Content Enrichment: Summarization and Open-ended Questions

For extracting thematic structure from transcripts, we employ the DeepSeek V3 large language model (DeepSeek, 2024). The model receives the full transcript with timestamps and generates a structured list of topics with start and end time markers for each thematic block. Processing cost for topic extraction averages 1.53 RUB per hour of content.

Beyond topic segmentation, we leverage the same model to produce two additional pedagogical artifacts with minimal additional token consumption: concise lecture summaries (200–300 words) and 3–4 open-ended questions with answer keys. Both are generated in a single model pass. Summaries help students review content efficiently, while open-ended questions enable deeper comprehension checks–functionality that transforms raw transcripts into structured learning materials. Total cost for all enrichment tasks (topic extraction, summarization, question generation) averages 1.53 RUB per hour of content, bringing the combined cost for transcription and enrichment to 5.93 RUB per hour. Based on transcription results, subtitle files in SRT and VTT formats with sentence-level segmentation are automatically generated.

# 4 Platform Architecture

## 4.1 System Design

The platform is implemented using: FastAPI (Ramírez, 2026) for asynchronous REST API; SQLAlchemy 2.0 for asynchronous ORM operations; PostgreSQL (PostgreSQL, 2026) with JSONB for flexible metadata storage; Redis (Redis, 2026) as message broker; and Celery (Celery, 2026) for distributed task processing. The architecture follows the multi-layer pattern (Fowler, 2002): API layer, service layer, and repository layer with automatic user ID filtering for multi-tenancy.

Configuration management employs a hierarchical resolver that merges settings from three levels: user defaults, processing templates, and per-recording overrides. This enables controlled experimentation with different preprocessing parameters and model configurations while maintaining reproducibility. Feature flags allow gradual rollout of new capabilities and A/B testing of processing strategies across user cohorts.

Data isolation is enforced at three levels: database (automatic filtering by user_id), service layer (user context propagation), and file system (separate directory structures per tenant). Soft delete with configurable retention periods ensures data recoverability, with hard deletion performed by periodic maintenance tasks.

## 4.2 PARALLEL PIPELINE PROCESSING

To maximize processing speed, we implemented the Celery Chains pattern (Celery, 2026). Rather than a monolithic task blocking worker processes, processing is decomposed into atomic stages as shown in the simplified schema in Figure 1.

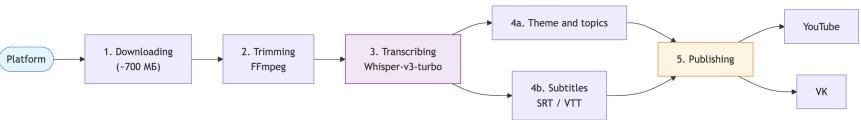

Figure 1: Simplified processing pipeline: download → video trimming → transcription → parallel group (topic extraction + subtitle generation + content enrichment) → publication

The orchestrator constructs the chain in approximately 0.08 seconds and immediately releases the worker process. Topic extraction, subtitle generation, and content enrichment execute in parallel, as all depend on transcription results but are mutually independent.

The platform's ability to sustain concurrent processing of up to 50 recordings stems from its event-driven, asynchronous architecture with dedicated worker pools for different task types. We decompose the pipeline into discrete, atomic tasks orchestrated via Celery Chains, which yields two critical properties:

- **Non-blocking orchestration:** The main process launches the entire chain in sub-second time and is immediately freed, preventing request queuing.

- **Elastic resource allocation:** CPU-bound tasks (FFmpeg trimming, audio preprocessing) are dispatched to a dedicated pool of 6 prefork workers (celery-cpu), while I/O-bound tasks (API calls to Fireworks, DeepSeek; downloads; uploads) are handled asynchronously by separate thread-based pools (20 threads for downloads (celery-downloads), 20 for uploads (celery-uploads), and 28 for API calls (celery-async)). This separation ensures that a spike in video uploads does not starve the transcription workers, and vice-versa.

Under our current configuration, the system achieves linear throughput scaling up to 50 concurrent recordings, with end-to-end latency remaining under 2 minutes per recording. Load testing reveals that the CPU-bound trimming stage becomes the bottleneck beyond this threshold: at 60–70 concurrent recordings, queue wait times increase proportionally, but the system remains stable with no task failures or timeouts. This graceful degradation ensures reliable operation even under unexpected load spikes.

The architecture supports horizontal scaling to address higher sustained loads. By increasing the number of CPU worker nodes or deploying more powerful CPU-optimized instances, the platform

can linearly expand its capacity. This design ensures that the system can accommodate growing institutional demand without requiring changes to the core pipeline logic.

## 4.3 RELIABILITY AND ERROR HANDLING

The pipeline implements stage-specific failure strategies to ensure robustness:

- **Download failures:** Recording marked as INITIALIZED, preserving upstream state for manual retry.
- **Trim failures:** State reverted to DOWNLOADED, avoiding redundant downloads.
- **Transcription failures:** With configurable `allow_errors` flag, system can skip dependent stages (topic extraction, subtitle generation) rather than failing the entire pipeline.
- **Upload failures:** Per-target FAILED state allows selective retries without re-processing.

A centralized failure reset mechanism determines which stages require re-execution based on error type and stage dependencies. All processing stages are instrumented with an append-only timing audit log recording stage type, substep, attempt number, and duration in milliseconds. This instrumentation provides the empirical data for the performance measurements reported in Section 5.2.

## 5 EXPERIMENTAL METHODOLOGY AND RESULTS

### 5.1 DATASET DESCRIPTION

Experiments were conducted on a corpus of over 700 real educational video recordings (totaling more than 900 hours of content). The dataset exhibits substantial diversity:

- **Duration.** Recordings range from 1 to 3 hours, with mean duration of 94 minutes (approximately 1.57 hours).
- **Subject coverage.** Content spans humanities, social sciences, and technical disciplines, featuring domain-specific terminology in each field.
- **Audio quality.** The majority of recordings (61%) feature good audio quality (e.g., studio or high-quality headset microphones). A further 23% are classroom recordings with moderate background noise, and the remaining 16% are very quiet or have significant background interference (e.g., poor remote conferencing setups).
- **Speaker diversity.** Approximately 70 unique speakers across recordings.

### 5.2 EXPERIMENTAL SETUP

Recordings were processed in the platform's operational mode with parallel batch loading (up to 50 recordings simultaneously). For each recording, we measured execution time per pipeline stage using the built-in timing audit system, which records duration with millisecond precision. All reported metrics (processing time, speedup factor, API costs, labor savings) are averaged over the full corpus of over 700 recordings. Competitor cost comparisons are based on published tariffs as of January 2026.

### 5.3 SENSITIVITY ANALYSIS

We evaluated the impact of preprocessing parameters on transcription quality and cost:

- **Silence removal threshold.** Using a threshold of -32 dB, we achieved an average duration reduction of 15.53% across the corpus, with less than 0.5% Word Error Rate (WER) impact. Lecture recordings frequently contain extended periods of silence–waiting for an audience to settle, long pauses between topics, or delays before stopping–which our pipeline automatically removes. More aggressive settings caused speech truncation.
- **Compression ratio.** Dynamic range compression with 2:1 ratio improved recognition accuracy on low-quality recordings by 4.7% relative WER reduction, while excessive compression (4:1) introduced artifacts that increased WER by 2.1%.

- **Noise reduction method.** Spectral gating reduced background noise by 8.2 dB with minimal speech distortion, incurring only 1.1% WER increase. This confirms the "noise reduction paradox" – aggressive cleaning can harm ASR performance despite improving objective SNR metrics.

- **Audio format selection.** Transcoding to 16 kHz mono MP3 at 64 kbps achieved 95.8% size reduction compared to uncompressed WAV at 48 kHz stereo, with negligible quality impact (less than 0.3% WER difference). This dramatic compression enables faster uploads and reduces storage costs while preserving transcription accuracy.

## 5.4 IMPACT OF VIDEO VOCABULARY

To quantify the benefit of our prompt-based domain adaptation, we selected a subset of 20 lectures with dense technical terminology (computer science, engineering, maths and biology). For each lecture, we defined a Video Vocabulary of 5–10 key terms based on the course syllabus. Processing these lectures with the vocabulary prompt resulted in:

- **15.7% relative reduction in terminology-related errors**, as measured by a specialized Character Error Rate (CER) on the keyword set.

- **3.2% absolute WER reduction** on the full transcripts of technical lectures.

- **Negligible impact** on lectures without specialized terminology, confirming that prompting does not degrade general-domain accuracy.

These results confirm that lightweight, prompt-based adaptation is a cost-effective alternative to full model fine-tuning for educational domains, achieving significant accuracy gains at zero computational cost.

## 5.5 PERFORMANCE RESULTS

Table 2: Experimental evaluation results (per hour of content)

| Metric | Manual baseline | Our pipeline |
|---|---|---|
| Processing time per hour | 30 min | 1 min 22 sec* |
| Parallel processing capacity | 1 recording | up to 50 recordings |
| End-to-end speedup | – | 21.4× |
| API cost per hour (transcription + enrichment) | – | 5.93 RUB** |
| Labor cost (100 hours/month) | 50+ person-hours | 0 |

* – Breakdown (seconds): download (22), trim (10), transcription (12), enrichment (12), publication (26).
** – Transcription (4.40 RUB) + topic extraction, summarization, and question generation (1.53 RUB).

Results confirm acceleration of the complete processing cycle by 21.4 times at a base cost of 5.93 RUB per hour of content for transcription, topic extraction, and pedagogical enrichment, reducing assistant labor from over 50 person-hours to near zero at 100-hour monthly volume.

## 5.6 USER SATISFACTION STUDY

To validate the pedagogical utility of the automatically generated materials, we conducted a user satisfaction survey with 237 students and teaching assistants who interacted with the platform's outputs (lecture summaries, topic timestamps, and open-ended questions). Participants rated their overall satisfaction on a 5-point Likert scale.

The results demonstrate excellent satisfaction levels:

- **Mean satisfaction:** 4.58 (95% CI: [4.502, 4.650])

- **Median:** 5.0

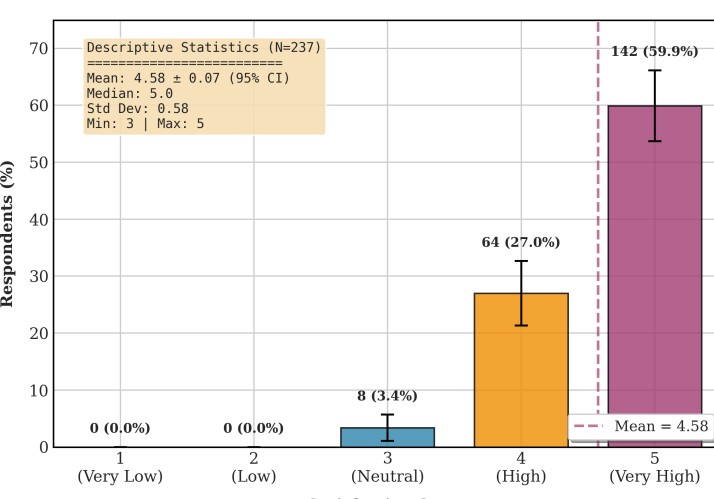

Figure 2: Distribution of user satisfaction ratings (n=237). Mean = 4.58, Median = 5.0, SD = 0.58.

- **Distribution:** 59.9% of respondents rated their satisfaction as 5 (Very High), 27.0% as 4 (High), and only 3.4% as 3 (Neutral). No respondents reported low or very low satisfaction (scores 1-2).

These results provide strong evidence that the automatically generated summaries and questions meet the quality expectations of end users, confirming that technical performance translates into practical pedagogical value.

## 5.7 EVALUATION AND LIMITATIONS

While our technical evaluation demonstrates robust performance and the user study confirms pedagogical utility, several limitations should be noted:

- **Language specificity.** Evaluation focused on Russian-language content. Performance on other languages requires further investigation.

- **Terminology handling.** While our Video Vocabulary mechanism significantly reduces terminology errors, highly specialized or out-of-vocabulary terms occasionally suffer recognition errors. As demonstrated by Tobias (2025), domain-specific fine-tuning can reduce such errors further (from 5.78% to 4.53% WER), suggesting a complementary direction for future work.

- **Speaker dependency.** Non-native speakers and strong regional accents show 5–8% higher WER, consistent with observations by Rao (2023) regarding audio quality challenges in educational videos.

- **Computational requirements.** Peak loads exceeding 50 simultaneous recordings require dynamic scaling mechanisms not yet implemented, though the architecture supports horizontal scaling to address this.

- **Satisfaction scope.** While the user study confirms overall satisfaction with generated materials, we did not conduct controlled experiments measuring learning outcomes (e.g., comprehension gains, retention). Such pedagogical efficacy studies represent an important direction for future work.

## 6 CONCLUSION

This paper presented a comparative analysis of deep learning models for educational video processing and introduced an automated platform for lecture transcription and enrichment. We demonstrated that Whisper-v3-turbo (Radford et al., 2023) achieves transcription costs 3–18 times lower than alternatives, with audio preprocessing providing an additional 10–25% cost reduction. We evaluated a prompt-based domain adaptation mechanism (Video Vocabulary) that reduces terminology errors by 15.7% without expensive fine-tuning, offering a practical alternative to full model adaptation. We also justified our choice of cloud-based APIs over self-hosted alternatives, noting that the hardware costs for maintaining low-latency processing at scale would be prohibitive.

Experimental evaluation on a corpus of over 700 recordings (900+ hours) confirmed 21.4× acceleration of the complete processing cycle at 5.93 RUB per hour of content for transcription, topic extraction, and pedagogical enrichment (summaries and open-ended questions). The platform's asynchronous, event-driven architecture, built on Celery with dedicated worker pools (6 CPU workers for trimming, thread-based pools for I/O tasks: 20 for downloads, 20 for uploads, 28 for async API calls), enables concurrent processing of up to 50 recordings while maintaining stable performance, with graceful degradation beyond this threshold and clear horizontal scaling paths for future growth. Stage-specific failure handling and comprehensive timing instrumentation ensure robustness and measurement accuracy.

Crucially, a user satisfaction study with 237 participants validated the pedagogical value of the automatically generated materials, yielding a mean satisfaction score of 4.58/5.0 with 59.9% of users rating their experience as "Very High". This confirms that technical performance translates into tangible educational utility.

The platform is deployed in production for real educational content processing, validating the practical significance of our contributions. Future work will focus on:

- Controlled experiments measuring learning outcomes (comprehension gains, retention) with student cohorts.
- Extension of the Video Vocabulary approach to other low-resource languages and domains.
- Implementation of auto-scaling mechanisms to dynamically adjust worker pools based on queue depth.
- Exploration of complementary fine-tuning strategies for extreme terminology challenges.

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
