# OpenReview forum: "Deep Learning for Educational Video Analysis: Benchmarking and Pipeline Optimization"
_mathai.club/MathAI/2026/Conference — MathAI 2026 Conference Submission_

### Official Review · Reviewer_4WNG · 2026-03-12
**A clear overview of deep learning applications in educational data mining, though the study lacks systematic methodology and deeper analytical insights.**

**Rating:** 6
**Confidence:** 4

**Review:**

Summary

This paper presents a scalable pipeline for processing educational lecture videos, including speech transcription, audio preprocessing, topic extraction, summarization, and question generation. The system benchmarks multiple ASR APIs and selects Whisper-v3-turbo as the most cost-efficient solution. Experiments on 700+ recordings (900+ hours) demonstrate 21.4× speedup compared to manual processing and a cost of 5.93 RUB per hour for transcription and enrichment. A user study with 237 participants reports high satisfaction (mean 4.58/5).

123_Deep_Learning_for_Educatio

Strengths

Large real-world dataset and production deployment.

Clear system architecture with scalable parallel processing.

Useful benchmarking of transcription API costs.

Practical domain adaptation via prompt-based “Video Vocabulary”.

User evaluation supporting educational usefulness.

Weaknesses

Limited methodological novelty; most contributions are engineering integration.

Cost comparisons are strong, but WER accuracy comparisons between models are limited.

Evaluation focuses only on Russian-language lectures, limiting generalization.

User study measures satisfaction rather than learning outcomes.

Overall Assessment

A well-written and practical systems paper demonstrating scalable AI infrastructure for educational video processing. While the algorithmic novelty is limited, the large-scale evaluation and production deployment make the work valuable.

---

### Official Review · Reviewer_dX1m · 2026-03-12
**Deep Learning for Educational Video Analysis: Benchmarking and Pipeline Optimization**

**Rating:** 7
**Confidence:** 5

**Review:**

This study focuses on the creation and optimization of an automated pipeline for transcribing and structuring educational videos.A comparative analysis of seven speech recognition (ASR) services was conducted on a corpus of 700+ lectures (900+ hours) in Russian. The Whisper-v3-turbo model, using Fireworks AI, was identified as the optimal solution in terms of price/performance (3-18 times cheaper than similar solutions).
  The system is built on FastAPI, Celery (for distributed processing), PostgreSQL, and Redis. The architecture supports parallel processing of up to 50 videos simultaneously (pages 4-5). The DeepSeek V3 model was used to extract topics by timecode and generate summary notes and quizzes (page 4).
   The novelty of this work lies in its comprehensive approach to optimizing not only the accuracy but also the cost and speed of an industrial solution. Unlike the works (Rao, 2023; Tobias, 2025), the study covers the entire cycle, from the raw video to the finished teaching aid, with a satisfaction assessment of real users (pp. 2, 7).
Conclusion: The system has been successfully implemented in production, confirming its practical value for educational institutions.

---

### Official Review · Reviewer_rDyZ · 2026-03-14
**Review of the article**

**Rating:** 3
**Confidence:** 4

**Review:**

## Section 1: Summary

The paper addresses the automation of educational video processing using deep learning methods. The authors compare seven commercial speech recognition APIs on a corpus of more than 700 lecture recordings totaling over 900 hours. Whisper-v3-turbo, deployed via Fireworks AI, is identified as the optimal solution for Russian-language content in terms of price-to-quality ratio. The authors propose audio preprocessing methods that reduce costs by 10–25% and a "Video Vocabulary" mechanism for domain adaptation through prompt engineering without fine-tuning. The platform, built on FastAPI, Celery, and PostgreSQL, enables parallel processing of up to 50 recordings and accelerates the full cycle by a factor of 21.4 compared with manual processing.

## Section 2: Strengths

The work is based on a real production corpus of more than 700 recordings, which significantly exceeds the scale of typical academic ASR studies. This lends the results a degree of practical relevance, although certain methodological limitations discussed below constrain the extent to which statistical significance can be claimed.

The proposed platform demonstrates careful engineering, including separation into CPU-bound and I/O-bound worker pools, a Celery Chains mechanism for parallel processing, and multi-level data isolation.

The end-to-end approach, integrating transcription optimization, audio preprocessing, domain adaptation, and generation of pedagogical materials into a single pipeline, distinguishes the paper from narrowly specialized ASR studies. The Video Vocabulary mechanism offers a lightweight alternative to computationally expensive fine-tuning, with a reported 15.7% relative reduction in terminology-related errors at zero training cost, though the statistical robustness of this result requires further validation.

The platform has been deployed in production, which supports its practical applicability. However, the scientific generalizability of the results is limited by the methodological issues outlined below.

## Section 3: Weaknesses and Limitations

### 3.1. Conflation of Models and Systems

The title claims a comparison of deep learning models, yet what is actually compared are commercial cloud-based APIs. A model has a specific architecture, fixed weights, and can be deployed locally with full control over inference. A cloud-based system is a complex product that includes preprocessing, chunking, postprocessing, infrastructure optimization, and potentially unknown model versions. The title is therefore misleading. Comparing cloud services constitutes a product comparison rather than a scientific comparison of architectures or methods.

### 3.2. Incomplete Coverage of Russian-Language ASR Models

Given the stated focus on the Russian language, the authors omit several high-quality Russian ASR models, including GigaAM 3 by Sber, T-one by T-Bank, Vosk by AlphaCephei, and community-developed models such as bond005/whisper-podlodka-turbo and Vikhrmodels/Borealis. Only the vanilla version of Whisper via Fireworks AI is considered, with no mention of Russian-language adaptations that consistently demonstrate higher recognition quality. This is a serious gap that calls into question the validity of the conclusion that the chosen solution is optimal.

### 3.3. Omission of Optimized Whisper Pipelines

The Related Work section fails to mention widely known optimized pipelines: Faster-Whisper (a CTranslate2-based optimization achieving approximately 4x inference speedup), WhisperX by Bain et al. (Interspeech 2023, addressing precise temporal synchronization through phoneme-level alignment), and Pisets (NAACL 2025, a robust system specifically designed for lectures). These works are directly relevant and their absence diminishes the claimed novelty and indicates an insufficiently thorough literature review.

### 3.4. Methodological Shortcomings of the User Satisfaction Study

The authors report a user satisfaction survey with 237 participants yielding a mean score of 4.58 on a 5-point Likert scale. However, the study does not meet accepted scientific standards for user studies.

The recruitment method, inclusion and exclusion criteria, target population size, response rate, and demographic characteristics of respondents are not specified, making it impossible to assess representativeness. The evaluation procedure is opaque: it is unclear how many materials each respondent evaluated, whether cross-evaluation was conducted, and whether any quality control measures such as attention checks were employed. Inter-rater reliability metrics are not reported.

The measurement instrument itself is not validated. It is unclear what "overall satisfaction" measures, whether a pilot study was conducted, or whether a validated scale such as the System Usability Scale was used.

The study is also vulnerable to several sources of bias. Self-selection bias is likely if respondents were volunteers from among active users. Social desirability bias may arise if students were reluctant to criticize a system introduced by their institution. Acquiescence bias is possible without a balanced scale. Finally, the absence of a human baseline makes it impossible to determine whether the reported score constitutes a good result.

In sum, the reported user study amounts to a post-hoc collection of satisfaction data without proper methodological design, and the authors' claim regarding pedagogical utility lacks adequate scientific support.

### 3.5. Reproducibility Concerns

No link to a code repository is provided. The corpus cannot be published for privacy reasons, but no synthetic dataset or detailed data format description is offered as an alternative. Preprocessing hyperparameters are only partially specified: a silence threshold and compression ratio are given, but the spectral gate parameters, voice activity detection algorithm, and encoding settings are not described. No example prompts for Video Vocabulary are provided, and the criteria for term selection are not explained. Standard deviations and confidence intervals are not reported for key metrics, and the claim of a 15.7% error reduction on 20 lectures lacks a formal significance test. The hardware used for experiments is not specified.

### 3.6. Insufficient Formal Rigor

The choice of preprocessing parameters is based on empirical tuning without theoretical justification or formal sensitivity analysis. The Video Vocabulary mechanism is described empirically without analysis of how the prompt influences token probability distributions, whether mode collapse or other artifacts might result, or what the optimal vocabulary size is and why. No computational complexity analysis of the pipeline is provided, and the claim of linear throughput scaling is not supported by formal analysis. For a venue such as MathAI, a stronger formal grounding would be expected.

### 3.7. Ethical Considerations

Voice data are biometrically identifiable, yet no re-identification risk assessment is discussed. The authors note a 5–8% higher word error rate for non-native speakers and speakers with strong regional accents, representing a potential source of bias in an educational context without discussed mitigation strategies. The claimed reduction in labor from over 50 person-hours to near zero raises questions about impact on employment of teaching assistants and support staff. The use of cloud-based APIs means data are transmitted to third parties, and it is unclear whether students have been informed of this.

## Section 4: Questions for the Authors

**Question 1.** Why does the comparative analysis consider cloud-based APIs rather than specific ASR models? How can the contribution of the model be separated from that of infrastructural optimization? Did the authors consider deploying models locally for a more controlled comparison?

**Question 2.** For what reason were Russian-language models such as GigaAM 3, T-one, Vosk, and Russian-language Whisper adaptations excluded from consideration? Could the authors conduct an additional comparison?

**Question 3.** Why does the Related Work section lack references to Faster-Whisper, WhisperX, and Pisets?

**Question 4.** Please describe in detail the methodology of the user satisfaction study: sample formation, respondent demographics, evaluation procedure, bias control measures, and inter-rater reliability.

**Question 5.** What is the statistical design of the Video Vocabulary study? Was the sample of 20 lectures selected randomly or purposively? Has a test of statistical significance been conducted with reported p-values and confidence intervals?

**Question 6.** How is the privacy of students' voice data ensured? Has a re-identification risk assessment been conducted? What is the informed consent protocol?

Satisfactory answers to Questions 1–3 and 4–5 are critical for evaluating the scientific soundness of the work. Unconvincing responses would support a recommendation to reject.

## Overall Recommendation: Reject

The paper demonstrates substantial engineering effort and has practical value for educational institutions. However, from the perspective of a scientific publication at the MathAI conference, the work exhibits fundamental methodological deficiencies.

First, the title promises a comparison of deep learning models, but what is actually compared are commercial cloud services. Second, the omission of Russian-language state-of-the-art models and optimized Whisper pipelines renders the conclusions insufficiently substantiated. Third, the user study lacks fundamental elements of scientific design and its results cannot be considered valid or reproducible. Fourth, the level of formal rigor is insufficient for a venue such as MathAI. Fifth, the absence of published code, incomplete hyperparameters, and missing statistical reporting do not meet accepted reproducibility standards.

For reconsideration, a thorough revision would be required: shifting the focus to a controlled comparison of specific ASR models with local deployment, including Russian-language models and Whisper adaptations, adding references to optimized pipelines, redesigning the user study with proper methodological rigor, ensuring reproducibility through published code and detailed statistics, adding formal analysis of the Video Vocabulary mechanism, and discussing ethical considerations.

In its current form, the paper is better suited to the format of an engineering report or technical white paper than to a peer-reviewed scientific conference publication.

---

### Decision · Program_Chairs · 2026-03-23

**Decision:**

Accept (Oral)

**Comment:**

Dear Author(s),

On behalf of the Program Committee of the International Conference on Mathematics of Artificial Intelligence (MathAI 2026), we are pleased to inform you that your paper has been accepted for an oral presentation at MathAI 2026.

Your paper was evaluated through a rigorous two-stage review process involving both automated screening and expert review by members of the Program Committee. The reviewers recognized the quality and contribution of your work.

Presentation details:

Format: Oral presentation (15–20 minutes + 5 minutes Q&A)
Mode: You may present either in person (offline) at the conference venue in Sirius, Russia, or remotely via Zoom. Please indicate your preferred mode when confirming your participation.
Conference dates: Marh 30 - April 3, 2026
Website: https://mathai.club
Next steps:

Please confirm your participation and presentation mode by replying to this email mathai.club@yandex.ru no later than March 15, 2026 18:00 Moscow time.
If you plan to attend in person, the organizing committee will provide accommodation details separately.
Please prepare your final camera-ready manuscript according to the formatting guidelines available at https://mathai.club and upload it to OpenReview by March 15, 2026 18:00 Moscow time.
Should you have any questions regarding the program, logistics, or your presentation slot, please do not hesitate to contact us.

We look forward to your contribution to MathAI 2026.

With kind regards,

MathAI 2026 Program Committee International Conference on Mathematics of Artificial Intelligence https://mathai.club OpenReview: https://openreview.net/group?id=mathai.club/MathAI/2026/Conference Telegram: https://t.me/MathAI_club Email: mathai.club@yandex.ru